# Building Evidence for Principles to Guide the Development of Products for Adults with Intellectual and Developmental Disabilities and Extreme Low Literacy—A Product Development Tool

**DOI:** 10.3390/healthcare11121742

**Published:** 2023-06-14

**Authors:** Linda Squiers, Molly M. Lynch, Sidney L. Holt, Aileen Rivell, Kathleen Walker, Stacy Robison, Elizabeth W. Mitchell, Alina L. Flores

**Affiliations:** 1RTI International, Research Triangle Park, NC 27709, USA; mlynch@rti.org (M.M.L.); sholt@rti.org (S.L.H.); arivell@rti.org (A.R.); 2CommunicateHealth, Rockville, MD 20850, USA; kathleen@communicatehealth.com (K.W.); stacy@communicatehealth.com (S.R.); 3Centers for Disease Control and Prevention, Atlanta, GA 30333, USA; bhm0@cdc.gov (E.W.M.); ail5@cdc.gov (A.L.F.)

**Keywords:** intellectual disabilities, developmental disabilities, low literacy, health communication, information needs

## Abstract

This article presented a new product development tool for adults with intellectual and developmental disabilities (IDD) developed by the Centers for Disease Control and Prevention (CDC). People with IDD who also have extreme low literacy (ELL) have unique communication needs; public health communicators often face challenges developing effective communication materials for this audience. To support CDC communication specialists with the development of communication products for adults with IDD/ELL, CDC, with its partners RTI International and CommunicateHealth, created a product development tool for this audience through literature review, expert input, and interviews with adults with IDD/ELL and caregivers of adults with IDD/ELL. To build evidence around the principles described in the tool, RTI conducted interviewer-administered surveys with 100 caregivers who support people with IDD/ELL. During the interviews, we presented caregivers with stimuli (portions of a communication product) that either did or did not apply a single principle and asked which would be easier for the person they support to understand. Across all 14 principles tested, the caregiver respondents indicated that the principle-based version would be easier for the person they support to understand compared with the non-principle-based version(s). These findings provide additional evidence to support the principles included in CDC’s Tool for Developing Products for People with IDD/ELL.

## 1. Introduction

During public health emergencies such as the COVID-19 pandemic, the Centers for Disease Control and Prevention (CDC) and other public health organizations must quickly develop communication messages, materials, and products to provide educational information to the public. CDC developed tools such as the clear communication index (CCI) to help ensure that most members of the public will understand public health messages [1]. However, people with intellectual or developmental disabilities (IDD), and especially those with very low literacy, may still struggle to understand messages that meet current plain language and clear communication standards. As a result, people with IDD often do not have the information they need during public health emergencies. Because the COVID-19 pandemic disproportionately affected people with IDD and other underlying conditions [2], CDC sought to make COVID-19 information more accessible to individuals with IDD/ELL. Developing information that factors in the literacy level of the primary audience is a constant consideration and challenge, and the need is even greater when developing information for people with IDD who also have extreme low literacy (ELL).

CDC defines an intellectual disability as a limitation to a person’s ability to learn at an expected level and function in daily life [3]. A systematic review found the prevalence of intellectual disabilities (IDs) in children to range from 11 to 13.4 per 1000 [4]. A developmental disability (DD) is defined as a limitation in physical, learning, language, or behavior areas [5]. About one in six children in the United States was diagnosed with a DD [6]. It is estimated that approximately 7.4 million people have IDD [7]; however, finding an exact prevalence is difficult because of the overlap in how ID and DD are defined and the broad range of severity in the impact of the disability [8].

The capabilities of people with IDD vary considerably. Some adults with IDD live on their own without assistance, whereas others receive support to perform activities of daily living. This support can be provided by either a paid or unpaid (generally a family member) direct care service provider or through full-time assistance in a congregate care setting. According to the American Association on Intellectual and Developmental Disabilities [9], most people in the United States with IDD live with their families and many need long-term caregiving support. Others live in community or small congregate care settings with individualized or small group support. Regardless of where they live or whether they receive care support, people with IDD need information to stay safe and healthy. This information needs to be provided in a way that is relevant, timely, easy to understand, and actionable.

Healthy People 2030 defines health literacy as “the degree to which individuals have the ability to find, understand, and use information and services to inform health-related decisions and actions for themselves and others” [10]. Most U.S. adults cannot understand health information in the way that it is usually presented [11]. Moreover, approximately 63 million U.S. adults have basic literacy, meaning they can only perform simple literacy tasks, while another 30 million have ELL or below basic skills [12]. This limited literacy can contribute to poor health outcomes and an inability to act on critical health recommendations. Most national surveys that examine literacy appear to exclude people who are unable to complete the surveys or assessments because of cognitive or language limitations. Therefore, it is unknown how many people with IDD also have ELL. However, according to a report of the National Adult Literacy Survey, people “with any type of disability, difficulty, or illness are especially vulnerable and more likely to perform at the lowest literacy levels” [12].

Although many public health organizations attempt to ensure that their information, messages, and materials are understandable and actionable, people with IDD who also have ELL (IDD/ELL), as well as others who experience ELL, may still have difficulty understanding public health messages. The CDC currently uses plain language and clear communication tools to support the development of materials for a variety of audiences, including the CCI, an evidence-based tool that can be used to develop new or assess existing public communication products for diverse audiences [1]. However, developing products for adults with IDD/ELL that are understandable, relevant, and usable can be challenging even for seasoned health communication specialists, health educators, and public health professionals. There are few standardized or evidence-based tools to guide material development for this audience; those available include the Guidelines for Minimizing the Complexity of Text and European Easy to Read Standards [13,14].

Because of the unique communication needs of people with IDD/ELL and the challenges public health communications staff faced developing effective materials for this audience, CDC, with its partners RTI International and CommunicateHealth, created a tool to help support CDC communication specialists with the development of communication products for adults with IDD/ELL. In the sections that follow, we describe the methods that we used to develop and test the tool.

## 2. Materials and Methods

### 2.1. CDC’s Product Development Tool

This stand-alone tool, created for CDC staff and contractors, was designed to guide product development for those who may not be familiar with the IDD/ELL audience. The tool included a user guide and a score sheet. The product development tool can be found at https://www.cdc.gov/ccindex/pdf/idd-ell-product-development-tool-508.pdf (accessed on 11 June 2023). The user guide outlined principles for developing communication products for adults with IDD/ELL and the score sheet enabled CDC staff and contractors to determine how well a product follows these principles.

The tool was composed of 27 principles. Each principle was worded as a question, and responses to each question assessed whether the principle was implemented: a “yes” response received 1 point, whereas a “no” response received 0 points (see Figure 1). For 10 items, a “not applicable” response option was available. Having principles in the form of a scored index allows an organization such as CDC to set its own standard for products developed for people with IDD/ELL. For CDC, a passing score is 90%.

#### 2.1.1. Methods to Create the Tool

To create the tool, we used a three-pronged approach. First, we conducted an environmental scan and a literature search to find existing best practices, principles, guidelines, standards, and recommendations for developing products for adults with IDD/ELL. Second, we obtained input from experts in health literacy and literacy on the draft tool and incorporated their feedback. Third, we conducted formative research with adults with IDD/ELL and their caregivers (i.e., family members or support providers for adults with IDD/ELL) to help identify their information needs and preferences and to explore the utility of draft principles in the tool. Altogether, we spoke to 36 adults with IDD/ELL, 54 caregivers, six health care providers, and three health literacy experts across four phases of testing. Table 1 displays the 27 principles in the tool and the 14 principles tested in this study.

#### 2.1.2. Methods to Test Principles in the Tool

Despite a review of the literature, a scan of relevant websites, and interviews with subject matter experts, we were unable to identify evidence-based principles or standards for communicating health information specific to people with IDD who also have ELL. Because there is such scant research on communication guidance for adults with IDD/ELL, it was imperative that we tested individual principles within the tool.

The draft tool included 29 principles and was organized by five domains: behavior, sentences, words and numbers, text layout, and visuals. While all principles included in the draft tool had some evidence that supported their inclusion, we wanted to strengthen the evidence base for 14 principles. We chose to test this subset of 14 principles because there was less available evidence—either from the current literature or our previous formative research findings—to support them.

To test the individual efficacy of the selected 14 principles, we created stimuli (i.e., part of a communication product) that either did or did not apply a single principle (see Figure 2). This strategy allowed us to isolate and test the effect of applying or not applying the principle. Specifically, we sought to learn whether respondents were more likely to select stimuli that apply the principles as easier for the person they support to understand compared with stimuli that do not apply the principles.

We selected caregivers of people with IDD/ELL as the respondents for the survey for several reasons. Caregivers play an important role in helping to communicate health information to adults with IDD/ELL. Previous studies found that caregivers who have high familiarity with the person they support provide accurate reports for children and adults with IDs [15,16], and patients with dementia [17]. When conducting our formative research on adults with IDD/ELL, we found that although many individuals with IDD may be able to participate in a virtual interview involving ratings by themselves, having ELL added an extra layer of communication challenges that were mitigated by engaging with the caregiver, as well. Our previous research showed that caregivers were essential in identifying communication recommendations that would facilitate use of products by adults with IDD/ELL and that they were able to serve as a proxy responder for the person with IDD/ELL as well as respond in their role as caregiver/communication intermediary. Caregivers were often better able to engage with and understand the feedback provided by the person with IDD/ELL [18].

### 2.2. Recruitment and Data Collection

The study sample of 100 caregiver respondents was recruited using a professional recruitment firm. To be eligible for the survey, respondents had to be aged 18 years or older; speak English; be able to participate in a virtual, 30 min interview using Zoom; and serve as a primary caregiver or support person for someone aged 15 years or older. In addition, the person the respondent supported had to have been diagnosed with an intellectual or developmental disability that was mild, moderate, or severe (those with profound levels were excluded) and have ELL.

To assess the severity of the supported person’s IDD, we asked caregivers to indicate how they or the supported person’s doctor described the severity of the intellectual disability using four categories: mild, moderate, severe, and profound. *Mild* described individuals with an IQ of 55 to 70 who were mostly self-sufficient with supports, may need additional time or instruction for basic life skills, but were able to be independent in other areas, and learn at about an elementary school level. *Moderate* described individuals with an IQ of 35 to 54 who may be able to communicate and have some independence in some areas; however, they need regular support for making decisions and need extended instruction and support for self-help skills. *Severe* generally described individuals with an IQ of 20 to 34 whose communication was very basic and limited and who required daily assistance for most self-care activities and need ongoing supervision for safety. *Profound* described individuals with an IQ of less than 20 who were dependent on others for all aspects of daily care. Caregivers who reported that the IDD of the person they supported was profound were not eligible for this study.

To obtain a general idea of the literacy level of the person with IDD, we asked caregivers to answer the following question: “What is the reading ability of the person you support? Choose which category best describes their reading ability”. Response options were as follows: (1) can read simple chapter books; (2) can read basic books with pictures; (3) can identify sight words; (4) can read common signs or pictures only; and (5) cannot read at all. We considered the person with IDD to have ELL if the caregiver said category 2, 3, or 4. Individuals whose caregivers chose categories 1 and 5 were ineligible for this study.

The professional recruitment firm sent invitations to its panel. Once potential respondents received the invitation, they were prompted to click a weblink to complete an online screener. Potential respondents who screened as eligible were: provided with more information about the study, asked if they were still interested in participating, and (if interested) asked demographic questions about themselves and the person they supported. Eligible respondents were then scheduled for a 30 min interviewer-administered survey.

We collected data in February and March 2022. Because COVID-19 was still prevalent during this time, we used Zoom screen-sharing technology to conduct the interviewer-administered survey. For each principle, a trained interviewer used PowerPoint slides to share (1) a stimulus that applied a single principle and (2) a stimulus that did not apply the principle. For two principles, respondents had three stimuli to select from: one that applied the principle and two that did not apply the principle. Using a structured guide, the interviewer then asked respondents to identify the difference between the two versions. Regardless of the response, the interviewer disclosed the difference between each version—for example, “In Version A, numerals are used to describe the number of shots needed. In Version B, the number of shots needed is spelled out”. This was to ensure that respondents were considering the difference being tested when asked which version would be easier for the person they support to understand and why. A notetaker entered responses into an Excel database. Interviews were also audio recorded.

### 2.3. Analysis

RTI used IBM SPSS Statistics 25 Armonk, USA to clean and prepare the data [19], conduct exploratory data analysis, and conduct descriptive statistics. We calculated frequencies and percentages for categorical variables and descriptive statistics, such as mean and standard deviations, for continuous variables to identify categories for recoding variables for descriptive and analytical purposes. We calculated frequencies and percentages for how often respondents selected stimuli as easier for the person they support to understand.

## 3. Results

### 3.1. Respondent Characteristics

Survey respondents were 100 caregivers of people with IDD/ELL. Table 2 presents their demographic characteristics. The caregiver respondents mirrored individuals with IDD/ELL in race and ethnicity, with most identifying as non-Hispanic White (68%) and some identifying as non-Hispanic Black or African American (25%). Additionally, the highest educational level of caregiver respondents varied, with 36% having a college degree and 19% having a graduate or professional degree. Other respondents reported attending college but no degree (19%), an associate’s degree (13%), a high school education or GED (8%), and attending some graduate or professional school (5%).

The caregiver respondents also provided information about the person they support. Respondents reported supporting a person with IDD/ELL across a range of ages, with most people with IDD/ELL being between 15 and 39 years old (68%) and the rest being older than 40 years old (32%). Respondents also indicated that 66% of people with IDD/ELL they provided care for identified as non-Hispanic White, while 24% identified as non-Hispanic Black or African American. Some people with IDD/ELL identified as Hispanic and White (8%), Hispanic Native Hawaiian or Pacific Islander (1%), and Hispanic with unidentified race (1%).

More than half of the caregiver respondents provided care for people with reported moderate IDD (62%), while some provided care for people with severe IDD (21%) or mild IDD (17%). More than half of the respondents said the person with IDD/ELL could read basic books with pictures (54%), whereas some people with IDD/ELL could only read common signs or pictures (25%) or could only identify sight words (20%). Most caregivers reported that the person with IDD/ELL did not ever read educational materials on their own, but rather reviewed them with a caregiver or other adult (82%). Characteristics of the people with IDD/ELL supported by respondents are detailed in Table 3.

### 3.2. How Often Caregivers and Respondents Selected the Version That Followed the Principle

Across all 14 principles tested, the caregiver respondents indicated that the principle-based version would be easier for the person they support to understand compared with the non-principle-based version(s). Two examples of principles and feedback received are provided below.

Principle: Does the product use one single main character to demonstrate all of the behavioral steps?


*“I think it would be confusing for someone with IDD to realize that it’s the same task and all the steps are for the same task, if it’s different people doing it.”*


Principle: Does the product use both text and visuals to illustrate each step or action?


*“If he’s not understanding the picture, there are some written cues that he could read through if he needed to.”*


For all but one principle, the percentage of respondents that selected the principle-based version ranged from 63% to 96%. For the one remaining principle (*22. Are all headers and titles informative statements?*), the principle-based version was the most often selected (39%); however, respondents were split between selecting the two non-principle-based options (30% and 29%).

Table 4 presents the percentage of the 100 respondents that selected the principle-based or non-principle-based version as easier for the person they support to understand. “Other” responses include cases where respondents did not indicate a preference for the principle- or non-principle-based version.

## 4. Discussion

Because people with IDD/ELL have unique communication needs, existing clear communication standards are not sufficient to create materials that are understandable and effective for this audience. The purpose of this study was to contribute to the evidence for specific principles within CDC’s Tool for Developing Products for Adults with IDD/ELL. Although we typically assess comprehension, access, and application of information that is presented in a product with the intended audience, doing so is difficult among adults with IDD/ELL because their cognitive abilities can vary considerably. Our previous research found that caregivers serve as communication intermediaries for people with IDD/ELL and are reliable proxy respondents given their knowledge of the capabilities and preferences of the person with IDD/ELL they support [18].

In this study, we tested principles that had minimal or mixed evidence of increasing comprehension among adults with IDD/ELL. To fill the gaps in evidence, we conducted an interviewer-administered survey of 100 caregivers of people with IDD/ELL to determine if the caregivers thought that stimuli that applied the principles would be easier for the person with IDD/ELL they support to understand compared with stimuli that did not apply the principles. The sample of caregivers was relatively diverse and the people with IDD/ELL they supported also ranged in age, severity of IDD, and reading level. Caregivers reported that few of the people with IDD/ELL they supported reviewed materials on their own, which indicates that most caregivers within the sample served as communication intermediaries.

For all 14 principles tested, we found that caregiver respondents selected the stimuli that applied the principles over the stimuli that did not. We also found almost consensus-level endorsement around several principles that provide clear guidance. Both principles we tested related to numeracy—Are whole numbers used? and Are all numbers shown as numerals rather than spelled out?—were selected by 90% or more of respondents as being more understandable. This finding highlights the importance of how and when numbers should be presented for this audience. We also found that using literal images (95%) and starting and finishing sentences on the same page (95%) were almost universally endorsed. The principle Are all headers and titles informative statements may have received less support (39%) because there were three stimuli rather than two and because a question format for headers and titles was commonly used and was familiar. The question format may be perceived as helping to the person with IDD/ELL know that the answer to the question will follow.

### Limitations

This study had several limitations. The biggest limitation to our study was that caregivers served as proxy responders for the person with IDD/ELL whom they supported. Given that caregivers often serve as information intermediaries, finding information resources and reviewing those resources with the person they support, we leveraged caregiver feedback in this survey as one perspective to triangulate with our initial formative interviews with adults with IDD/ELL. Although adults with IDD/ELL were included in the formative research process, in future research, we hope to obtain more direct feedback from adults with IDD/ELL, when we can do so safely outside of a virtual setting. We acknowledge that the use of proxy responders should never be used as the single source of data for people with intellectual or development disabilities, especially when pertaining to internal thoughts or feelings. Because we used convenience sampling, our sample may not be representative of caregivers who support adults with IDD/ELL. In addition, the severity of the IDD and reading level were reported by the caregiver and not directly assessed.

## 5. Conclusions

Despite these limitations, this study provided additional evidence to support the principles included in CDC’s Tool for Developing Products for People with IDD/ELL. The tool included a user guide that provided background about adults with IDD/ELL and described why developing communication products for adults with IDD/ELL required a different approach and way of thinking that went beyond plain language principles and that making edits to existing products developed for the public will not suffice. Communicators should define their audience, identify the most important information this audience needs to know, create clear behavioral recommendations, and consider a format that this audience can access and enjoys using (e.g., social stories, videos, and interactive materials). The user guide showed both the correct and incorrect way to apply each principle through examples and illustrations. In addition to the scored principles, the user guide described practices that were unique to developing products for adults with IDD/ELL, including the following:Cut anything that is not essential;Write short, straightforward sentences with common, literal words;Keep images literal and realistic;Use social stories to give step-by-step instructions;Choose respectful, empowering language, including honoring your audience’s preference for identity-first or person-first language;Avoid “othering” people with disabilities; andTreat adults like adults.

## Figures and Tables

**Figure 1 healthcare-11-01742-f001:**
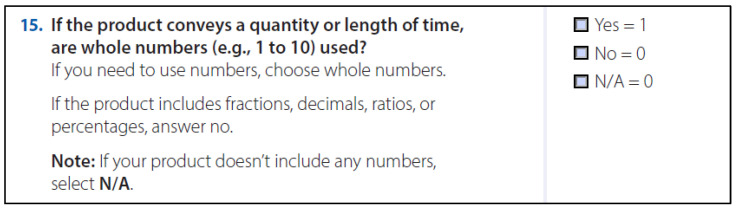
Sample Principle with Scoring Instructions.

**Figure 2 healthcare-11-01742-f002:**
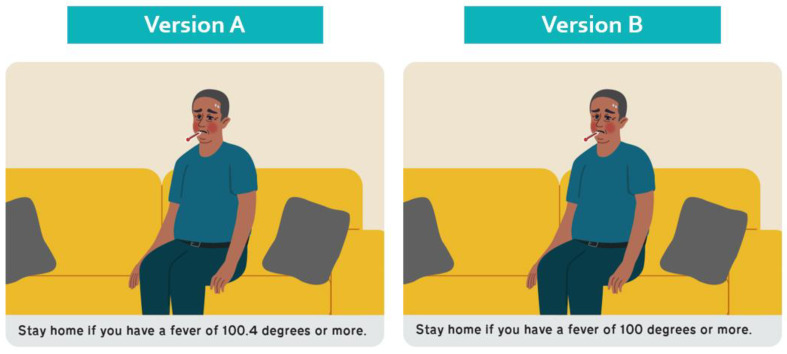
Sample Stimuli Used in Testing: If the product conveys a quantity or length of time, are whole numbers (e.g., 1 to 10) used?

**Table 1 healthcare-11-01742-t001:** Product Development Tool and Items Tested.

		Response Options			Principles Tested in This Study ^2^
Principles		Yes	No	NA ^1^	
A. Behavior	1. Does the product include only 1 focused behavioral recommendation?	1	0		Yes
	2. Is the behavioral recommendation stated more than 1 time?	1	0	0	Yes
	3. Does the product show how to perform the behavioral recommendation by breaking the behavior down into a series of single steps or actions?	1	0	0	
	4. Does the product convey only 1 idea or behavioral step per page or screen?	1	0		Yes
	5. Does the product use both text and visuals to illustrate each step or action?	1	0		Yes
	6. Does the product use one single main character to demonstrate all of the behavioral steps?	1	0		Yes
B. Sentences	7. Does each sentence focus on 1 key point?	1	0		
	8. Do all (or almost all) sentences, headers, and titles use 10 or fewer words?	1	0		Yes
	9. Are all sentences in the active voice?	1	0		
	10. Are all headers and titles informative statements?	1	0	0	Yes
	11. Is a consistent tense used throughout the product?	1	0		
C. Words and numbers	12. Does the product always use words the primary audience understands?	1	0		
	13. Does the product clearly define any need-to-know jargon terms using familiar words?	1	0	0	
	14. Does the product always use the same word for the same concept?	1	0		Yes
	15. If the product conveys a quantity or length of time, are whole numbers (e.g., 1 to 10) used?	1	0	0	Yes
	16. Are all numbers shown as numerals rather than spelled out (e.g., 1, not “one”)?	1	0	0	Yes
D. Text layout	17. Is text left aligned?	1	0	0	
	18. If the product is on more than 1 page, is the text formatted the same way on each page?	1	0	0	
	19. Does every sentence finish on the same page it starts?	1	0		Yes
	20. Does the product use a single sans-serif font?	1	0		
	21. Does the product always use a 14-point or larger font size?	1	0		
	22. Is the product free from italicized or underlined words?	1	0		
E. Visuals	23. Is only 1 visual included on each page or screen?	1	0	0	
	24. Does each visual have no more than 1 to 2 lines of corresponding text that describes what is happening in the visual?	1	0		
	25. Is text supporting the visual positioned directly below the visual?	1	0		Yes
	26. Are all visuals literal images of the item or action rather than abstract symbols?	1	0		Yes
	27. When using an illustration of a person, are at least minimal facial features included (eyes, nose, and mouth)?	1	0	0	Yes

^1^ Not applicable. ^2^ Principles not tested had sufficient evidence or were established clear communication principles.

**Table 2 healthcare-11-01742-t002:** Respondent Characteristics (*n* = 100).

Demographic	Caregiver %
Age (years)	
18–29	5
30–39	17
40–49	23
50–59	34
60–69	15
70 or older	6
Race/ethnicity	
Non-Hispanic White	68
Non-Hispanic Black or African American	25
Hispanic White	6
Hispanic unknown race *	1
Language other than English spoken in the home	
Spanish	5
American Sign Language (ASL)	2
None	93
Education level	
High school graduate or GED	8
Some college, no degree	19
Associate’s degree	13
College graduate	36
At least some graduate or professional school	5
Graduate or professional degree	19

* One respondent who identified as Hispanic did not disclose their race.

**Table 3 healthcare-11-01742-t003:** Characteristics of the Person with IDD/ELL (*n* = 100).

Demographic	Caregiver %
Age (years)	
15–19	25
20–29	26
30–39	17
40–49	12
50–59	10
60 or older	10
Race/ethnicity	
Non-Hispanic White	66
Non-Hispanic Black or African American	24
Hispanic Native Hawaiian or Other Pacific Islander	1
Hispanic White	8
Hispanic unknown race *	1
Severity of IDD	
Mild	17
Moderate	62
Severe	21
Reading level	
Can read only common signs or pictures	25
Can identify only sight words	20
Can read basic books with pictures	54
Missing	1
Person with IDD ever reviews materials on their own	
No	82
Yes	18

* One respondent who indicated the person with IDD/ELL identified as Hispanic did not disclose their race.

**Table 4 healthcare-11-01742-t004:** Percentage of Respondents That Selected Each Version of the Stimuli (*n* = 100) as More Understandable.

Domain	Principle	Principle-Based Version %	Non-Principle-Based Version %	2nd Non-Principle-Based Version %	Other %
Behavior	Does the product include only 1 focused behavioral recommendation?	69	30	NA	1
	Is the behavioral recommendation stated more than 1 time?	89	10	NA	1
	Does the product convey only 1 idea or behavioral step per page or screen?	80	16	NA	4
	Does the product use both text and visuals to illustrate each step or action? *	84	15	0	1
	Does the product use one single main character to demonstrate all of the behavioral steps?	84	10	NA	6
Sentences	Do all (or almost all) sentences, headers, and titles use 10 or fewer words?	73	22	NA	5
	Are all headers and titles informative statements? **	39	30	29	2
Words and numbers	Does the product always use the same word for the same concept?	77	21	NA	2
	If the product conveys a quantity or length of time, are whole numbers (e.g., 1 to 10) used?	90	5	NA	5
	Are all numbers shown as numerals rather than spelled out (e.g., 1, not “one”)?	94	2	NA	4
Text layout	Does every sentence finish on the same page it starts?	96	3	NA	1
Visuals/images	Is text supporting the visual positioned directly below the visual?	63	33	NA	4
	Are all visuals literal images of the item or action rather than abstract symbols?	95	5	NA	0
	When using an illustration of a person, are at least minimal facial features included (eyes, nose, and mouth)?	89	7	NA	4

* Non-principle-based includes 2 versions: text only and visuals only to illustrate each step or action—15 respondents selected the visuals-only non-principle-based version and no respondents selected the text-only non-principle-based version. ** Non-principle-based includes 2 versions: headers and titles as simple statements and questions—30 respondents selected the simple statement non-principle-based version and 29 respondents selected the question non-principle-based version. NA = not applicable.

## Data Availability

Not applicable.

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
