# Peer review of "Building Evidence for Principles to Guide the Development of Products for Adults with Intellectual and Developmental Disabilities and Extreme Low Literacy—A Product Development Tool"

_healthcare, 2023, doi:10.3390/healthcare11121742_

Round 1

Reviewer 1 Report

The proposal of this article is interesting since it is regularly investigated in relation to people who are within normality, however, the study of people with disabilities and low literacy makes a special differentiation. The observations offered are the following:

a). The title of the document proposes "Develop products" and in reality the research does not develop but rather applies the effectiveness of a device in specific conditions. In addition, it is proposed to delete the expression: "CDC's How".

b). In a very serious way, the differential classification of the participants in the study is striking. For example, Non-Hispanic White, Non-Hispanic Black or African American (they are not really synonyms), Hispanic White, Hispanic Unknown race (this is not understood). This classification, in addition to being considered incorrect, shows too much bias in the classification of the participants or in their choice. I think this is the most critical point of the study since it never defines the type of sampling and the way in which the sample was selected.

c). It is necessary to redefine the results of the study, either by specifying results with caregivers or proxy caregivers and without them. If caregivers participated in all situations, it is necessary to specify this in the results.

d). Within the limitations, he speaks of a small sample. This is a non-scientific statement, it may be small or not, it simply must be representative and duly selected. This aspect must be corrected by explaining the exact way in which the sample was chosen, even in cases of populations with such particular characteristics as is the case in this article, case studies are used and the sample could even be smaller.

e). In matters of format, Table 2 is considered unnecessary, it can be sent to an annex or it can only be explained in its structure, since Table 4 repeats the criteria without numbering.

Reviewer 2 Report

I only reviewed the statistical aspects of the manuscript. The paper presents descriptive statistics to present its argument. There are no inferential analyses nor is there a control group comparison. I believe that there is a limitation of sampling by only including those subjects that have access to the Zoom system. Also, the SPSS software should be cited.

Reviewer 3 Report

None.

Reviewer 4 Report

Congratulations to the authors for this very important study.

The abstract provides a concise summary of the article, highlighting the key points and findings. Overall, the abstract effectively summarizes the purpose, methodology, and main findings of the article in a clear and concise manner. It provides the reader with an overview of the study and its contribution to the field of developing communication products for adults with IDD/ELL.

The introduction provides background information about the challenges of developing effective communication materials for people with intellectual or developmental disabilities (IDD) and extreme low literacy (ELL). It mentions that existing plain language and clear communication standards may not be sufficient for this audience. The introduction also highlights the importance of providing accessible and actionable information to individuals with IDD/ELL, considering their unique communication needs. Overall, the introduction provides a comprehensive overview of the challenges and the need for effective communication materials for individuals with IDD/ELL. It sets the stage for the development and testing of the tool, which is further described in subsequent sections.

The methodology is robust and comprehensive. The development of the tool involved a three-pronged approach, including an environmental scan and literature search, input from experts in health literacy and literacy, and formative research with adults with IDD/ELL and their caregivers.

The environmental scan and literature search aimed to identify existing best practices, principles, guidelines, standards, and recommendations for developing products for adults with IDD/ELL. This step ensured that the tool was built upon existing knowledge and evidence.

Input from experts in health literacy and literacy was obtained to incorporate their feedback and expertise into the development of the tool. This consultation with experts adds credibility and strengthens the validity of the tool.

The formative research involved engaging with adults with IDD/ELL and their caregivers to identify their information needs and preferences and to explore the utility of the draft principles in the tool. This direct engagement with the target audience and their caregivers is crucial for understanding their perspectives and ensuring that the tool aligns with their needs.

The methodology also mentions testing a subset of 14 principles to strengthen the evidence base for these specific principles. This targeted testing allows for a more focused evaluation and provides an opportunity to gather data on the effectiveness of these principles.

The use of caregivers as respondents for the survey is justified based on previous studies highlighting their important role in communicating health information to adults with IDD/ELL. Caregivers' high familiarity with the individuals they support and their ability to serve as proxy responders and communication intermediaries make them suitable respondents for the study.

Overall, the methodology demonstrates a rigorous approach that combines literature review, expert input, and direct engagement with the target audience and caregivers. This multi-phase approach enhances the credibility and validity of the tool's development process.

The limitations of the study and the need for future research to obtain more direct feedback from adults with IDD/ELL. It also states that the study provides additional evidence to support the principles included in CDC's Tool for Developing Products for People with IDD/ELL.

The results section of the excerpt provides information on the characteristics of the caregivers and the individuals they support, as well as the caregivers' feedback on the tested principles.

Regarding the characteristics of the individuals with IDD/ELL supported by the respondents, most fell within the age range of 15 to 39 years (68%), while the remaining were older than 40 years (32%). The majority of the respondents reported supporting individuals who identified as non-Hispanic White (66%) or non-Hispanic Black or African American (24%). Some individuals identified as Hispanic and White, Hispanic Native Hawaiian or Pacific Islander, or Hispanic with an unidentified race.

In terms of the severity of IDD, most caregivers reported supporting individuals with moderate IDD (62%), followed by severe IDD (21%) and mild IDD (17%). When it came to reading ability, over half of the respondents stated that the individuals with IDD/ELL they supported could read basic books with pictures (54%), while others could only read common signs or pictures (25%) or identify sight words (20%). The majority of caregivers mentioned that the individuals did not read educational materials on their own but reviewed them with a caregiver or another adult (82%).

The section then discusses the caregiver respondents' feedback on the tested principles. Across all 14 principles, the majority of respondents indicated that the principle-based version of the communication product would be easier for the person they support to understand compared to the non-principle-based version(s). Two examples of the feedback provided by caregivers for specific principles are given, demonstrating their perspectives on the relevance and effectiveness of the principles.

The results suggest that the tested principles in the tool received positive feedback from caregivers, with the majority perceiving them as enhancing understanding for individuals with IDD/ELL. This provides support for the effectiveness of the principles included in CDC's Tool for Developing Products for People with IDD/ELL.

Overall, the conclusion provides a summary of the study's findings and emphasizes the importance of the developed tool and user guide in guiding the development of effective communication products for adults with IDD/ELL. It provides practical recommendations for communicators to ensure that their materials are accessible, relevant, and respectful to this specific audience.

Well done.

Round 2

Reviewer 1 Report

I am concerned that the observations made previously were not addressed or justified, could you please consider them:

b). In a very serious way, the differential classification of the participants in the study is striking. For example, Non-Hispanic White, Non-Hispanic Black or African American (they are not really synonyms), Hispanic White, Hispanic Unknown race (this is not understood). This classification, in addition to being considered incorrect, shows too much bias in the classification of the participants or in their choice. I think this is the most critical point of the study since it never defines the type of sampling and the way in which the sample was selected.

c). It is necessary to redefine the results of the study, either by specifying results with caregivers or proxy caregivers and without them. If caregivers participated in all situations, it is necessary to specify this in the results.

d). Within the limitations, he speaks of a small sample. This is a non-scientific statement, it may be small or not, it simply must be representative and duly selected. This aspect must be corrected by explaining the exact way in which the sample was chosen, even in cases of populations with such particular characteristics as is the case in this article, case studies are used and the sample could even be smaller.

e). In matters of format, Table 2 is considered unnecessary, it can be sent to an annex or it can only be explained in its structure, since Table 4 repeats the criteria without numbering.

Author Response

Reviewer 1

I am concerned that the observations made previously were not addressed or justified, could you please consider them:

b). In a very serious way, the differential classification of the participants in the study is striking. For example, Non-Hispanic White, Non-Hispanic Black or African American (they are not really synonyms), Hispanic White, Hispanic Unknown race (this is not understood). This classification, in addition to being considered incorrect, shows too much bias in the classification of the participants or in their choice. I think this is the most critical point of the study since it never defines the type of sampling and the way in which the sample was selected.-

AUTHORS' RESPONSE: Per editor’s feedback we suggest leaving as is. These are census designations that are routinely used in reporting demographics for subjects in the United States. Race is reported in addition to ethnicity (i.e., Hispanic). See: https://www.census.gov/programs-surveys/decennial-census/decade/2020/planning-management/release/faqs-race-ethnicity.html

c). It is necessary to redefine the results of the study, either by specifying results with caregivers or proxy caregivers and without them. If caregivers participated in all situations, it is necessary to specify this in the results.

AUTHORS' RESPONSE: Caregivers were the participants in this study.  See this paragraph in the manuscript:  "We selected caregivers of people with IDD/ELL as the respondents for the survey for several reasons. Caregivers play an important role in helping to communicate health information to adults with IDD/ELL. Previous studies have found that caregivers who have high familiarity with the person they support provide accurate reports for children and adults with IDs [15, 16], and patients with dementia [17]. When conducting our formative research on adults with IDD/ELL, we found that although many individuals with IDD may be able to participate in a virtual interview involving ratings by themselves, having ELL added an extra layer of communication challenges that were mitigated by engaging with the caregiver, as well. Our previous research showed that caregivers were essential in identifying communication recommendations that would facilitate use of products by adults with IDD/ELL and that they were able to serve as a proxy responder for the person with IDD/ELL as well as respond in their role as caregiver/communication intermediary. Caregivers were often better able to engage with and understand the feedback provided by the person with IDD/ELL [18]."

d). Within the limitations, he speaks of a small sample. This is a non-scientific statement, it may be small or not, it simply must be representative and duly selected. This aspect must be corrected by explaining the exact way in which the sample was chosen, even in cases of populations with such particular characteristics as is the case in this article, case studies are used and the sample could even be smaller.

AUTHORS' RESPONSE: To address the issue of sampling, we added this statement:   "Because we used convenience sampling, our sample may not be representative of caregivers who support adults with IDD/ELL." 

e). In matters of format, Table 2 is considered unnecessary, it can be sent to an annex or it can only be explained in its structure, since Table 4 repeats the criteria without numbering.

AUTHORS' RESPONSE: Table 2 is necessary because we tested only 14 of the principles, not all 27 of them.  This was indicated in the manuscript